# Understanding older adults' adoption of facial recognition payment: An integrated model of TAM and UXT

**Mingzhou Liu**[1], **Caixia Wang**[1,2], **Jing Hu**[1]*

**1** School of Mechanical Engineering, Hefei University of Technology, Hefei, China, **2** North Minzu University, Yinchuan, China

* IEHFUT@163.com

## Abstract

The dual challenges of digitalization and population aging highlight the need to improve the well-being of older adults. Facial recognition payment (FRP), an innovative "face-to-pay" technology, has gained widespread adoption in commercial applications due to its convenience. However, existing research has primarily focused on younger users, leaving a significant gap in understanding FRP adoption and user experience among older adults. To address this gap, this study integrates the Technology Acceptance Model (TAM) with User Experience Theory (UXT) to develop a comprehensive framework for examining FRP adoption among older users. By incorporating both cognitive and affective dimensions, the proposed model systematically investigates key determinants of FRP acceptance. Data were collected through a questionnaire survey of 387 older adults in China, and hypotheses were tested using structural equation modeling (SEM). The findings reveal that TAM-based factors, including perceived convenience, facilitating conditions, and technology anxiety, significantly influence perceived ease of use (PEOU), perceived usefulness (PU), and user attitudes, with perceived convenience identified as the most influential determinant. From a UXT perspective, perceived enjoyment, personal innovation, and privacy risk also play a crucial role in FRP adoption. Specifically, personal innovation and perceived enjoyment positively affect user attitudes, whereas privacy risk poses a barrier to adoption. Additionally, while technology self-efficacy does not directly impact user attitudes, it indirectly enhances PEOU and PU by increasing personal innovation. By integrating TAM and UXT, this study advances the understanding of older adults' technology acceptance behavior and offers theoretical and practical implications for the development of age-friendly FRP systems. The findings provide valuable guidance for technology developers, policymakers, and stakeholders in designing user-centered payment technologies that enhance usability, security, and overall user experience, ultimately improving the quality of life for aging populations.

**Data availability statement:** All data underlying the findings described in our manuscript are fully available without restriction. The data can be found in the public dataset on figshare. The URL of the dataset is: https://doi.org/10.6084/m9.figshare.28774835, and the DOI is 10.6084/m9.figshare.28774835.

**Funding:** The author(s) received no specific funding for this work.

**Competing interests:** The authors have declared that no competing interests exist.

# 1. Introduction

Population aging has become a pressing global issue. World Bank projections indicate that by the year 2100, the proportion of individuals aged 65 and older will rise to 22.49% of the total global population, amounting to approximately 2.52 billion people [1]. In China, this demographic trend is particularly significant, with the elderly population expected to reach 190 million, representing 13.5% of the national population [2]. As this transformation progresses, promoting the well-being of older adults has become a vital societal imperative. The rapid evolution of technology offers innovative pathways to address the unique challenges faced by aging populations [3]. Among these advancements, facial recognition payment (FRP) technology is increasingly viewed as a potential solution to enhance convenience and accessibility for senior users.

With the widespread adoption of digital payment technologies, facial recognition payment (FRP) has emerged as an innovative and contactless payment method. China has played a pioneering role in the commercialization and implementation of facial recognition technology. On September 1, 2017, Alipay launched "Face Pay" in a pilot program at a KPRO restaurant in Hangzhou, marking the world's first trial of FRP in a retail setting. Since then, FRP has gained widespread acceptance and adoption, becoming a key component of both online and offline payment ecosystems. However, while FRP has been extensively deployed across various industries, there is limited research on its usability and acceptance among older adults.

Older adults face unique challenges when adopting new technologies. Prior studies have emphasized that technological innovations can act as catalysts for promoting rehabilitation and improving adaptability, effectively sustaining the functional capacities of older individuals facing age-related challenges [4,5]. From a consumer perspective, FRP offers potential advantages for elderly users, such as eliminating the need for remembering passwords and reducing physical interaction with devices. However, issues related to technology anxiety, technological self-efficacy and perceived convenience remain underexplored in the context of aging populations.

Although FRP provides notable benefits, its adoption among elderly users is still in its early stages. Existing research has primarily focused on younger users [6,7], with limited studies examining how older adults perceive and interact with this technology. Therefore, it is crucial to thoroughly investigate the attitudes and acceptance of elderly users towards FRP. Based on the aforementioned research deficiencies, this study develops an FRP adoption framework tailored for elderly users based on the Technology Acceptance Model (TAM) and User Experience Theory (UXT). By considering both the characteristics of elderly users and usage dimensions, this framework identifies the key factors influencing FRP adoption among the elderly and examines their impact on users' adoption intentions. The study aims to bridge existing research deficiencies, provide theoretical support for the age-friendly optimization of smart payment technologies, and promote the widespread adoption of FRP among aging populations, ultimately enhancing payment convenience and user experience.

## 2. Literature review

Facial recognition payment technology has garnered substantial scholarly attention due to its innovative approach to payment systems. Some studies have conducted certain explorations from the dimension of the theoretical framework. For instance, Lee utilized the Stressor-Strain-Outcome (S-S-O) framework to elucidate the factors contributing to consumer resistance toward facial recognition mobile payment services [8]. Shiau integrated the belief-attitude-intention (B-A-I) model with an extended TOE-I framework to investigate how perceived convenience, perceived social influence, trust, and satisfaction impact consumer acceptance of facial recognition technology [9]. Jin et al., employing Herzbergs motivation-hygiene theory, examined the impact of motivational and hygiene factors on the acceptance and resistance to mobile facial recognition payment services. Their findings highlight the positive influence of motivational factors on acceptance and the significant effects of hygiene factors on resistance [10]. Liu et al., drawing on privacy calculus theory and innovation resistance theory, constructed a research model for facial recognition payment (FRP) and investigated the privacy concerns among Chinese users [11]. Hwang integrated the Perceived Risk Theory and the Extended Theory of Planned Behavior to conduct a research on the behavioral intentions and switching intentions of South Korean and American consumers regarding the adoption of facial recognition technology in restaurant payment scenarios [12]. Meanwhile, most studies, based on the Technology Acceptance Model (TAM), have extensively explored the factors influencing users' adoption of facial recognition payment (FRP). For instance, a study conducted by researchers at the University of Macau identified a set of pivotal factors influencing the adoption of facial recognition payment systems. These determinants include perceived enjoyment, facilitating conditions, personal innovation, coupon availability, perceived ease of use (PEOU), perceived usefulness (PU), and users' overall attitudes toward the technology. They also noted significant gender disparities in adoption rates [13]. Fatemeh examined security concerns related to adversarial attacks in facial recognition systems [14]. Lai et al. conducted a study to explore the public's familiarity, trust, and attitudes toward different scenarios involving facial recognition technology. Their findings led to the development of a public perception model that categorizes these scenarios into four types [15]. Li et al. analyzed the effects of perceived vulnerability, perceived security, and perceived responsiveness on trust in facial recognition payment systems from privacy and security perspectives [16]. Furthermore, researchers have investigated additional factors affecting consumer adoption of facial recognition payment technologies, including social influence [17], hedonic shopping value [18], technology fear and perceived complexity [19], visibility and social image concerns [20], personal information protection capabilities, negative media exposure [21], perceived novelty [22], self-efficacy [23], and social influence [24].

Focusing on elderly users, the aging process is accompanied by a progressive decline in physiological functions and corresponding psychological changes. Research indicates that technological interventions can facilitate rehabilitation and enhance adaptability among elderly individuals [25,26]. Significant progress has been made in understanding the acceptance of facial recognition payment (FRP) technology within this demographic. Lee et al. reported that elderly users prioritize system reliability, ease of operation, and a sense of security when engaging with FRP systems [27]. Furthermore, Lyu et al. identified privacy concerns as a critical determinant of adoption intention. Their findings suggest that when elderly users perceive a high risk of data breaches, their willingness to use FRP decreases significantly [28]. Additionally, Lim et al. introduced the Artificial Autonomy Theory, which posits that a highly intelligent payment system capable of autonomously performing security verifications and issuing risk alerts can enhance user trust and promote sustained engagement [29]. From a cognitive perspective, Mahler and Murphy demonstrated that optimizing the interaction interface, reducing operational complexity, and providing clear visual feedback significantly improve cognitive fluency, thereby increasing elderly users' acceptance of FRP technology [30]. Similarly, Yulianti and Ghina highlighted that the intuitiveness of the payment interface and the comprehensiveness of user guidance play a pivotal role in shaping perceived ease of use (PEOU). Their study revealed that when FRP systems offer clear operational instructions and minimize unnecessary interactions, the cognitive load on elderly users is reduced, facilitating greater adoption [31]. Beyond

cognitive determinants, cultural factors exert a substantial influence on FRP acceptance among elderly users. Empirical evidence suggests notable cross-national variations in adoption behavior: Chinese users tend to prioritize system controllability and reliability, whereas their American counterparts place greater emphasis on privacy protection and data security. Korean users are predominantly influenced by functional and hedonic innovation motivations, while American users are more driven by social innovation factors [32,33]. Furthermore, emotional factors exert a stronger impact on Korean users, whereas convenience and social influence play a more significant role for American users [34]. These findings underscore the necessity of tailoring FRP implementation strategies to align with the cultural and contextual preferences of different user groups.

From a theoretical perspective, the Technology Acceptance Model (TAM) posits that Perceived Usefulness (PU) and Perceived Ease of Use (PEOU) are the primary determinants of users' acceptance of new technologies. However, TAM predominantly emphasizes rational decision-making processes, with limited consideration of affective factors and contextual influences on user behavior. This limitation is particularly salient for elderly users, whose technology adoption is influenced by multidimensional factors, including a sense of security, cognitive load, and trust formation [30]. As a result, there has been a growing academic emphasis on examining elderly users' technology acceptance from a User Experience (UX) perspective. For instance, Hornbæk and Hertzum investigated the intersection of TAM and UX models, highlighting the significance of experiential components in human-computer interaction. Their findings indicate that user experience exerts a greater influence on technology acceptance than PU and PEOU, particularly through perceived hedonic value and attitudinal formation [35]. Similarly, Xia et al. explored how online experiences shape users' perceptions of destination imagery within the TAM framework, revealing that, beyond technical considerations, emotional and cognitive factors significantly impact elderly users' technology acceptance [36]. Additionally, Cho et al. investigated the use of a UX-integrated TAM model in the context of autonomous driving, showing that UX-related factors, such as trust, safety, and emotional satisfaction, play a crucial role in shaping behavioral intentions. Their study further revealed that as the level of automation increases, UX and user acceptance (UX/UA) scores demonstrate significant variations, which ultimately impact adoption decisions [37].

Despite these advancements, existing research on elderly users' acceptance of Facial Recognition Payment (FRP) exhibits several limitations. First, most studies have predominantly focused on the initial adoption phase, neglecting factors influencing continued usage behavior. However, previous research suggests that user stickiness, perceived risk, and habit formation are crucial determinants of sustained engagement with technology [30]. Second, although TAM and User Experience Theory (UXT) are conceptually interconnected, extant studies often apply these frameworks in isolation, rather than integrating them into a comprehensive theoretical model. There remains a notable gap in research attempting to synthesize TAM and UXT to provide a more holistic understanding of the behavioral mechanisms underlying elderly users' adoption of FRP.

Facial recognition payment technology is particularly beneficial for elderly users. Firstly, in terms of convenience, older adults often experience declines in executive control within working memory, affecting their functional status in daily activities [38]. Facial recognition payment allows elderly individuals to effortlessly complete transactions without the need for passwords, card swipes, or handheld devices, offering unparalleled convenience compared to other technological systems [39,40]. Secondly, from the perspective of human-computer interaction experience, older adults often view digital technologies as challenging to understand and operate [41,42], which may lead to psychological obstacles, including diminished confidence in their abilities and heightened apprehension regarding computer usage [43,44], mistrust of technology adoption [45], and frustration [46]. Facial recognition payment simplifies the transaction process by requiring users only to scan their face, which is inherently simpler for older adults compared to other interactive methods [39,40].

## 3. Theoretical framework and hypothesis development

This research seeks to explore elderly users' acceptance of and inclination to adopt facial recognition payment systems. To build a robust theoretical foundation and identify critical influencing factors, a comprehensive literature review was

conducted. Key terms, including "Technology Acceptance Model (TAM)," "user experience," "facial recognition payment," and "elderly users" were applied to search prominent academic databases such as PubMed, IEEE Xplore, and Web of Science. This systematic approach aimed to identify well-established theoretical frameworks and empirically validated constructs pertinent to the research scope.

### 3.1 Theoretical Framework: Technology Acceptance Model (TAM) and User Experience Theory (UXT)

Among the theoretical models for examining user acceptance of technology, the Technology Acceptance Model (TAM), first introduced by Davis [47], remains one of the most widely recognized due to its emphasis on perceived ease of use and perceived usefulness as primary determinants of adoption behavior. As an extension of the Theory of Reasoned Action (TRA), TAM has undergone numerous refinements. For instance, TAM2 incorporated constructs related to social influence [48], while TAM3 further expanded its scope by integrating factors such as perceived enjoyment and user experience [49]. Moreover, Chen et al. [50] introduced the Senior Technology Acceptance Model (STAM), an adaptation of TAM tailored to older adults, incorporating additional elements such as self-efficacy, technology anxiety, and facilitating conditions to better explain their adoption behaviors. Similarly, the Unified Theory of Acceptance and Use of Technology (UTAUT), formulated by Venkatesh et al. [51], extends TAM by integrating constructs like social influence and facilitating conditions, offering a broader perspective on technology acceptance across various demographic groups.

A thorough review of the literature [52] emphasizes the central role of the Technology Acceptance Model (TAM) as a widely recognized and influential framework for understanding technology acceptance among older adults. The model has been widely applied across various domains, including intelligent systems, digital communication tools, healthcare technologies, and personal devices, demonstrating its substantial explanatory power and adaptability. These studies collectively underscore TAM's ability to identify and elucidate the key factors influencing the adoption of technologies by older adults, reinforcing its position as a foundational model in the field of technology acceptance research [53–64].

The core concept of User Experience is inherently user-centered, emphasizing individuals' overall perception and subjective responses when interacting with a product, system, or service. According to ISO 9241-210, UX is defined as the subjective experience users develop while using or anticipating the use of a product, system, or service [65]. However, its scope extends beyond this definition. The International Organization for Standardization further elaborates that UX encompasses the entire lifecycle of product or service usage, including pre-use expectations, real-time interactions, and post-use reflections [66]. It incorporates multiple dimensions, including emotional states, cognitive processes, preferences, perceptions, physiological and psychological responses, behavioral patterns, and a sense of achievement. Consequently, UX is recognized as a multidimensional construct that integrates cognitive, affective, and behavioral factors. Hassenzahl proposed that UX comprises two fundamental dimensions: Pragmatic Quality and Hedonic Quality. Pragmatic Quality pertains to the practicality and functional aspects of a product, encompassing usability, operability, and efficiency [67]. In contrast, Hedonic Quality pertains to the experiential and emotional aspects, including pleasure, satisfaction, and aesthetic appeal. These dimensions collectively shape users' perceptions and influence their interaction with technology.

In the context of facial recognition payment (FRP) for older adults, a comprehensive investigation into the cognitive and emotional determinants of adoption and sustained use is of significant theoretical and practical importance. Recent studies have begun integrating the Technology Acceptance Model (TAM) with User Experience Theory (UXT) to establish a more holistic framework for understanding the acceptance and engagement of older users with emerging payment technologies. **Table 1** presents a comprehensive synthesis of existing research on the acceptance of facial recognition payment (FRP) technology, highlighting key gaps in the literature. The Technology Acceptance Model (TAM) has been extensively utilized in prior studies [13,24,68–70], demonstrating its robustness in elucidating user adoption behaviors. However, most research has predominantly centered on younger populations, leaving a significant gap in empirical studies specifically examining the acceptance of FRP technology among older adults.

**Table 1. Selected summary of prior research on the adoption and acceptance of FRP.**

| Study Reference | Title of the Literature | Theoretical model | Factors considered | Sample & Data in Studies | Research Gaps |
|---|---|---|---|---|---|
| [13] | Service transformation under industry 4.0: Investigating acceptance of facial recognition payment through an extended technology acceptance model | TAM | perceived enjoyment, facilitating conditions, personal innovativeness, coupon availability, perceived ease of use, perceived usefulness, users' attitude | The study surveyed 247 participants, with 53% aged 21–30 and 34% aged 31–40. Most had used facial recognition payment for over six months. | The research focuses on the technology acceptance of facial recognition payment (FRP) among young users aged 21–40, neglecting the elderly user group |
| [20] | Factors Affecting the Use of Facial-Recognition Payment: An Example of Chinese Consumers | TAM | security, visibility, expected effort, social image, perceived usefulness, openness characteristic, usage intention | The survey resulted in 299 valid responses, and148 respondents were aged 21–30 (49.7% of the total). A significant portion were involved in business activities | The study centers on users aged 21–30, without addressing the acceptance of facial recognition payment technology among elderly users |
| [30] | Risk of desirable user experiences: insights from those who create, facilitate and accept mobile payments | Fluency Theory + User Experience Design | aesthetics, the need for a simplified experience, sensory elements that replicate familiar visual, audio, and haptic stimuli | Research data were gathered via semi – structured, face-to-face interviews with 12 participants. | The study has not systematically explored how factors such as cognitive load, trust, security, and feedback mechanisms influence the payment behavior of elderly users. |
| [35] | Technology acceptance and user experience: A review of the experiential component in HCI | TAM + UXT | perceived enjoyment, attitude, perceived usefulness, perceived ease of use, psychological needs, negative emotions | 37 papers in the overlap between TAM and UX models to explore the experiential component of human--computer interactions | Most reviewed studies aren't linked to specific use cases. This bypasses tasks as an explanatory variable and weakens the accurate measurement of experiences, which change moment by moment. |
| [37] | Technology acceptance modeling based on user experience for autonomous vehicles | TAM + UXT | Performance Expectancy (PE), Social Influence (SI), Perceived Safety (PS), Anxiety (AX), Trust (T), Affective Satisfaction (AS), Behavioral Intention (BI) | The experiment designed scenarios based on the four automation stages of autonomous vehicles. A total of 68 male and female participants took part in this experiment, and the acceptance assessment was conducted using a driving simulator. | The research concentrated on the technology acceptance among young users, yet overlooked elderly users. Additionally, the scope of user experience factors was confined solely to affective satisfaction. As such, the research remains inadequately comprehensive. |
| [71] | "Paying with my face" – Understanding users' adoption and privacy concerns of facial recognition payment | TAM | privacy concerns, power usage, attitudes, intentions of adopting FRP. | An online survey on cloud research was conducted on the Amazon MTurk platform, involving 164 highly qualified micro-workers. All participants are residents of the United States | The user group in the study is predominantly from the United States, with no consideration of elderly users in China |
| [72] | Investigating the technology acceptance model, image congruence and cultural differences in facial recognition payment adoption | TAM | perceived usefulness, perceived ease of use, attitude, actual self-image congruence | The data were collected via an online survey from 342 South Korean and 353 American consumers who patronized a restaurant within a three-month timeframe | The study focuses on users from the United States and South Korea, with a lack of analysis on age-related differences in the influencing factors |
| [73] | An integrative model of facial recognition check-in technology adoption intention: the perspective of hotel guests in Singapore | TAM | privacy concern, perceived risk, institutional trust, perceived benefits, attitudes | The survey data were collected from 374 guests who participated in an online survey across eight hotels in Singapore | The study focuses on users from Singapore and institutional environments, lacking applicability to broader facial recognition usage scenarios. Additionally, the research does not analyze age-related differences |

*(Continued)*

**Table 1.** (Continued)

| Study Reference | Title of the Literature | Theoretical model | Factors considered | Sample & Data in Studies | Research Gaps |
|---|---|---|---|---|---|
| [74] | Factors Affecting Intention of Consumers in Using Face Recognition Payment in Offline Markets: An Acceptance Model for Future Payment Service | TAM + UTAUT | attitude, subjective norm, perceived behavioral control, personal norm, cultural difference | Data was obtained via an online survey, involving 345 participants from Korea and 338 from the United States. | The user group in the research is centered on participants from the United States and South Korea, and the analysis does not account for age differences in the influencing factors |
| [75] | Exploring biometric identification in FinTech applications based on the modified TAM | TAM | perceived privacy, perceived trust, Perceived usefulness, Perceived ease of use, | A paper questionnaire survey was conducted among 264 users aged 22–33 who frequently utilize financial technology applications for cross-border remittances, electronic payments, student loans, insurance, and online investments | The study focuses on young users aged 22–33, without considering the impact of elderly users and age factors on the technology acceptance of voice recognition assistants, and lacks an analysis of age-related differences in the influencing factors |
| [76] | Understanding user acceptance of QR code mobile payment systems in Turkey: An extended TAM | TAM | perceived trust (PT), perceived compatibility (PC), perceived usefulness (PU), Perceived ease of use (PEOU) | The research data was obtained from 485 QR code MPS users in Turkey using an online survey | The study concentrates on users in Turkey, with the influencing factors lacking an analysis of age differences. |

Moreover, while some progress has been made in integrating TAM with User Experience Theory (UXT) [30,35–37], no systematic investigations have been identified that concurrently employ these two frameworks to comprehensively analyze FRP technology acceptance among older adults. This research gap underscores the need for further exploration of the interplay between technological and experiential factors in shaping adoption behaviors within this demographic.

Building upon the previous analysis, this study develops a research framework that integrates the Technology Acceptance Model (TAM) with User Experience Theory (UXT), incorporating seven supplementary variables. This extension aims to address the limitations of the traditional TAM framework in examining the acceptance of facial recognition payment (FRP) technology among older adults. The selection of variables is informed by three primary criteria: established theoretical frameworks [30,35,47,48], empirical evidence from prior research [10,16,36,37], and an understanding of the physiological and psychological characteristics of older adults [41–46]. Specifically, within the TAM framework, four additional constructs are introduced: Perceived Convenience (PC), Facilitating Conditions (FC), Technology Anxiety (TA), and Technology Self-Efficacy (TSE). From the user experience perspective, three supplementary constructs are integrated: Perceived Enjoyment (PE), Personal Innovation (PI), and Privacy Risk (PR). The hypothesized relationships among these factors are depicted in **Fig 1**. This study aims to provide a comprehensive analysis of the factors influencing the acceptance and adoption of FRP technology among elderly users, examining both technological and experiential dimensions. By offering valuable insights and practical recommendations, the study seeks to enhance the alignment of FRP technology with the specific needs of older adults, thereby improving its usability and adoption rate. Each construct is defined with relevant scholarly references, and the theoretical foundation underpinning the proposed causal relationships is further elaborated in the subsequent sections.

### 3.2. Perceived ease of use, perceived usefulness, attitude, behavioral intention

The Technology Acceptance Model (TAM) identifies Perceived Usefulness (PU) and Perceived Ease of Use (PEOU) as critical determinants influencing the acceptance and utilization of technology [77]. PU is conceptualized as the extent

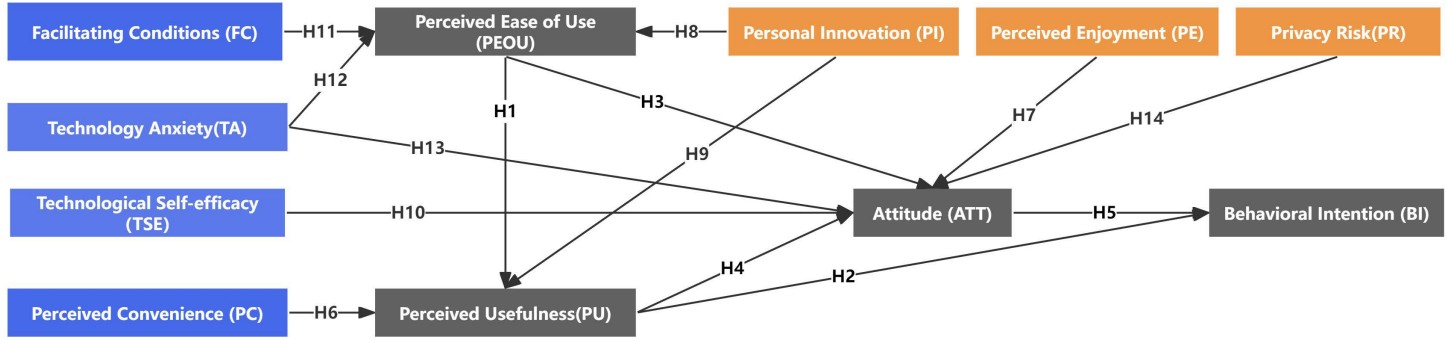

**Fig 1. Conceptual framework illustrating the research model and proposed relationships between variables.**

to which individuals perceive that a given technology enhances their performance. In the context of facial recognition payment systems, PU represents the perceived convenience and practical benefits that older adults associate with task completion facilitated by the technology [78,79]. PEOU, on the other hand, refers to the perception that using a technology requires minimal effort [77]. For facial recognition payment systems, this ease of use is exemplified by the system's reliance on facial authentication, eliminating the need for physical interaction, thereby offering a seamless user experience. TAM postulates that both PU and PEOU are significant predictors of Behavioral Intention (BI), with a notable interdependence between these constructs. Moreover, the model suggests that Attitude (ATT) toward the technology exerts a direct influence on BI, wherein a favorable attitude strengthens users' intentions to adopt the technology.

From the perspective of User Experience Theory (UXT), ease of use extends beyond merely reducing user effort; it also plays a crucial role in fostering a positive user experience by enhancing interaction efficiency and alleviating cognitive load. According to Hassenzahl's user experience model [67], intuitive and effortless interactions contribute to a more satisfying and engaging experience, which is particularly significant for older adults who may encounter cognitive and motor difficulties when engaging with digital technology. A positive attitude is typically shaped by both the hedonic and pragmatic aspects of the experience [66]. When a system is both functional (high Perceived Usefulness, PU) and enjoyable or frustration-free (high Perceived Ease of Use, PEOU), users are more likely to develop a favorable attitude toward it, consequently increasing their intention to adopt the technology. Based on these theoretical foundations, the study puts forward the following hypotheses:

H1: Perceived ease of use (PEOU) positively influences perceived usefulness (PU).

H2: Perceived usefulness (PU) positively influences behavioral intention (BI).

H3: Perceived ease of use (PEOU) positively influences attitude (ATT).

H4: Perceived usefulness (PU) positively influences behavioral intention (BI).

H5: Attitude (ATT) positively influences behavioral intention (BI).

The remaining seven exogenous constructs are detailed in the subsequent section.

### 3.3. Perceived convenience

Aging naturally results in a deterioration of sensory, perceptual, motor, and cognitive functions [38], which creates difficulties when engaging with devices that rely on conventional graphical user interfaces (GUIs) [78,80]. Brown emphasizes that convenience is defined by five key factors: accessibility (products/services should be conveniently located), timeliness (available when needed), usability (user-friendly interface), transaction (multiple financial channels for purchasing), and execution (decision-making convenience) [81]. Therefore, FRP is advantageous as it provides convenient service

at suitable times and places while being user-friendly. Compared to conventional methods like fingerprint or numerical/diagram passwords [82], FRP offers superior convenience. As a contactless payment option, paying solely through facial recognition offers customers an engaging experience without physical contact [83]. FRP has been implemented in various offline retail checkout systems and online platforms such as Alipay and online banking applications, enabling elderly individuals to benefit from convenient human-computer interaction anytime, anywhere. Considering the interactive advantage of facial recognition payment (FRP), which removes the necessity for physical interaction with a user interface, we propose that the convenience provided by this approach substantially shapes users' perceptions of its usefulness. Therefore, we propose the following hypothesis:

H6: Perceived Convenience (PC) positively influences perceived usefulness (PU).

### 3.4. Perceived enjoyment

Perceived enjoyment (PE) is recognized as a key factor influencing technology acceptance and adoption [84,85]. Hedonic motivation, defined as the pleasure or satisfaction gained from engaging with technology, plays a significant role in this process [86]. Numerous studies have emphasized the importance of hedonic drivers, such as enjoyment and playfulness, in shaping users' attitudes toward adopting new technologies [87–89]. When users find enjoyment in utilizing a technology, they may perceive it as more intuitive, even if it involves underlying complexities [90]. Research on mobile services [91] suggests that users who derive enjoyment from a service are more inclined to adopt it positively. Enjoyment can influence users' intentions to adopt technology either directly or indirectly through psychological mechanisms tied to their beliefs [92]. Additionally, Celik [90] found that enjoyment significantly enhances users' perceptions of Perceived Ease of Use (PEOU) and their willingness to engage in online shopping. From the user experience perspective, Perceived Enjoyment (PE) plays a pivotal role in facilitating the adoption of new technologies among older adults [66]. It not only enhances emotional engagement by reducing cognitive load and improving interaction efficiency but also contributes to increased overall satisfaction, thereby fostering sustained usage intentions. Hornbæk and Hertzum's study further underscores that the influence of perceived enjoyment on user attitudes often surpasses that of perceived usefulness and ease of use [35]. This highlights the critical importance of enjoyable experiences in shaping older adults' acceptance of new technologies, significantly enhancing their adoption intentions.

Within the consumer context, PE is recognized as a crucial component influencing technology acceptance and usage [86,93]. Facial recognition technology, as an innovative approach, holds appeal for customers exploring this new payment method. It offers an interactive experience that enhances shopping enjoyment by reducing wait times and improving service efficiency [94]. This novel method presents a simplified and enjoyable payment experience, potentially enhancing customers' acceptance and intention to adopt it. Based on these studies, the following hypothesis is proposed:

H7. Perceived Enjoyment (PE) positively influences attitude (ATT).

### 3.5. Personal innovation

Personal innovation (PI) can be defined as an individual's propensity to explore and experiment with emerging technologies [95]. This disposition significantly influences consumers' comprehension of and attitudes toward the utility and user-friendliness of new technologies [96]. According to the International Organization for Standardization [66], user experience encompasses multiple dimensions, including emotional states, cognitive processes, and individual preferences. Individuals with higher levels of personal innovation exhibit greater openness and optimism toward emerging technologies, making them more likely to engage with novel and innovative products or services. This propensity fosters a proactive approach to technological exploration, which in turn positively influences their intention to adopt new technologies. The indirect impact of personal innovation on the inclination to adopt Facial Recognition Payment (FRP) is increasingly apparent. For instance, Ciftci et al. [97] found no direct correlation between personal innovation and the intention to use facial recognition technology in quick-service restaurant environments. Similarly, The study by Palash et al. focuses

 

on consumer behavior regarding facial recognition payments (FRP) in offline markets, underscoring the critical role of personal innovation in shaping users' attitudes toward adopting new payment technologies [98]. Morosan underscored personal innovation as a pivotal determinant of perceived ease of use, which subsequently predicts adoption intentions for FRP in dining contexts [99]. Kim and Forsythe proposed that personal innovation directly impacts perceptions of ease of use (PEOU) and usefulness (PU) in the context of adopting product virtualization technology [100]. Moreover, it serves as a predictor of customers' cognitive attitudes toward embracing and applying information technologies [101]. In the absence of personal traits such as innovation, consumer attitudes and purchasing intentions may tend to reflect habitual choices among similar products [102]. Therefore, innovation emerges as a key predictor of consumers' motivation and readiness to explore novel concepts and technologies [100]. Consumers with a strong preference for novelty tend to have a positive outlook on emerging technologies and are more inclined to engage with innovative products, services, and experiences [103]. Currently, facial recognition technology remains relatively nascent among many older Chinese consumers. Those exhibiting higher levels of innovation are more inclined to embrace such innovations with enthusiasm [104]. The empirical findings from these studies underscore the pivotal role of personal innovation in shaping consumer perceptions of ease of use, usefulness, and overall attitudes toward technology adoption. Based on the aforementioned research, the following hypotheses are proposed:

H8. Personal innovation (PI) positively influences perceived ease of use (PEOU).

H9. Personal innovation (PI) positively influences perceived usefulness (PU).

### 3.6. Technological self-efficacy

Self-efficacy, a cornerstone concept in social cognitive theory, refers to individuals' assessments of their capability to execute specific actions [105]. In the context of technology applications, self-efficacy reflects individuals' confidence in possessing the necessary expertise and skills to effectively utilize technology [106]. Empirical research consistently underscores the pivotal role of self-efficacy in facilitating the adoption and proficient use of information technology [107,108]. Drawing upon social learning theory [105], older adults' confidence in their capacity to perform unfamiliar tasks, such as using computers or navigating the Internet, emerges as a critical determinant of their success in these domains. Investigations into Internet usage further highlight the extensive impact of self-efficacy, demonstrating its strong associations with constructs such as perceived ease of use (PEOU), perceived usefulness (PU) [109], behavioral intention (BI) [110], and overall patterns of Internet engagement [111]. This study defines technological self-efficacy as individuals' perceptions or judgments of their ability to effectively utilize Facial Recognition Technology (FRT). Researchers have integrated self-efficacy across various technological domains, including computers [112–114], the internet [115,116], mobile technologies [117–119], and robotics [120,121]. Self-efficacy plays a pivotal role in shaping individuals' behavioral decisions and the level of effort they invest in adopting and using technologies [122]. Prior literature consistently indicates that individuals with higher self-efficacy maintain positive attitudes and exhibit greater openness to adopting new technologies, driven by their confidence in mastering technological skills [123–126]. Individuals with elevated self-efficacy typically exhibit a positive outlook and display greater receptiveness toward the adoption of new technologies. This tendency reflects their confidence in mastering and effectively utilizing these innovations [127]. Building upon the aforementioned research, the following hypothesis is proposed:

H10: Technological self-efficacy (TSE) positively influences attitude (ATT).

### 3.7. Facilitating conditions

Facilitating conditions refer to customers' perceptions regarding the availability of essential resources and support needed to carry out a specific task [86,128]. It also encompasses motivational factors and environmental barriers that shape an individual's perception of the ease or difficulty associated with a task [129]. Derived from Venkatesh et al.'s [130] Unified Theory of Acceptance and Use of Technology (UTAUT), facilitating conditions are fundamental components within

the UTAUT model. Venkatesh et al. [130] posit that these conditions influence user behavior rather than just behavioral intentions, encompassing accessibility to necessary resources and support for technology use. Insufficient assistance, information, and resources can impede the adoption of web-based technologies [131]. Facilitating conditions are essential for users who heavily rely on technical support and resources when adopting new technologies. These conditions can significantly enhance users' attitudes [132,133] and increase their intention to use the technology [134].

Facial recognition payment methods typically require integration with other technologies or systems, such as linking to specific digital payment platforms or bank accounts (commonly WeChat Pay and Alipay in China). Given its novelty, older consumers may benefit from additional technical guidance, particularly during the initial adoption phases. Zhong et al. underscore the significance of facilitating conditions as an external variable influencing Perceived Ease of Use (PEOU), Perceived Usefulness (PU), and behavioral intentions [13]. This highlights the critical role of adequate resources or support in mitigating technological adoption challenges. Consequently, facilitating conditions are considered an independent variable crucial for assessing consumers' intentions to adopt facial recognition payment systems. Based on the above analysis, the following hypotheses are proposed:

H11: Facilitating Conditions (FC) positively influences perceived ease of use (PEOU).

### 3.8. Technology anxiety

Anxiety refers to a range of negative emotions, including fear, sadness, and tension, that arise in response to stressful circumstances [135]. In the realm of information and communication technology (ICT), technology anxiety denotes users' apprehension and unease about the potential adverse outcomes associated with specific technological applications [136,137]. Extensive research has empirically demonstrated that anxiety negatively influences users' intentions [138–140]. For instance, McFarland and Hamilton, utilizing the Technology Acceptance Model (TAM), demonstrated that computer anxiety exerts a significant influence on users' intentions to engage with computers [141]. Similarly, Lu and Su identified anxiety as a barrier to the adoption of innovative systems, revealing a negative association with customers' intentions to adopt mobile phones [142]. Further studies, including those by Celik and Yesilyurt [143] and Patil et al. [144], have consistently highlighted the detrimental effects of anxiety on users' attitudes toward technology. Compeau and Higgins [145] posited that individuals are inclined to avoid activities that induce anxiety. Those experiencing heightened levels of technological anxiety often exhibit reduced confidence in navigating technology, leaving them more prone to feelings of vulnerability or unease when confronted with technological challenges [146]. Against this backdrop, the current study aims to examine the relationship between technological anxiety and attitudes toward adopting facial recognition payment (FPR), thereby proposing the following hypotheses:

H12: Technology Anxiety (TA) negatively influences perceived ease of use (PEOU).

H13: Technology Anxiety (TA) negatively influences Attitude (ATT).

### 3.9. Privacy risk

Privacy risk (PR) refers to individuals' concerns regarding potential privacy breaches when disclosing information to specific external entities [147,148]. Such concerns profoundly shape individuals' trust in external entities and their approaches to establishing personal boundaries within their external surroundings [149]. User experience encompasses both perception and emotional state [66]. The influence of privacy risk is largely driven by users' subjective perceptions rather than objective realities, often eliciting feelings of anxiety, unease, or concern [28]. These emotional responses can significantly shape users' attitudes toward technology and their subsequent adoption behaviors. In the context of facial recognition payment (FRP), users frequently divulge private information, such as contact details and demographic data, during mobile transactions, whether consciously or unconsciously. Additionally, FRP services necessitate users to submit facial information stored in sensitive databases [150]. Consequently, users' privacy attitudes often drive their resistance to adopting FRP [11,21]. Research indicates that privacy risk diminishes users' perceived value of mobile payment services [151].

Recent studies underscore privacy risk as a critical factor in increasing user resistance to FRP, particularly when users perceive heightened privacy risks associated with this technology [152,153]. Thus, we hypothesize the following:

H14: Privacy Risk (PR) negatively influences Attitude (ATT).

## 4. Research methodology

### 4.1. Data acquisition

A structured survey questionnaire was designed and conducted to assess the proposed hypotheses and evaluate the conceptual framework. While acknowledging the inherent limitations of such a survey, it remains a valid instrument for this study. Given the cognitive and technological challenges that elderly populations may face, careful consideration was given to the questionnaire's design to ensure its suitability. In an effort to reduce potential biases, a more refined sampling method was employed, combining both online and offline data collection approaches. Online surveys, commonly used in prior research on user adoption of FRP [71–76], were used to gather data from participants, primarily university faculty members and retired employees aged 55 and above. However, recognizing the unique needs of elderly participants, additional face-to-face interviews were included. This hybrid approach ensured that elderly respondents could independently complete the questionnaire, thereby minimizing the risk of participants providing inaccurate responses for compensation. The data collection focused on individuals aged 55 and older. While 60 is commonly used as the operational definition of older adults, we also included female participants aged 55-60 in consideration of China's official retirement policy, where the retirement age is set at 60 for men and 55 for women. Given that many retired individuals in this age group exhibit behavioral patterns and technology adoption characteristics similar to those aged 60 and above, their inclusion allows for a more comprehensive understanding of elderly users' attitudes toward FRP. To ensure that all participants had a basic understanding of FRP systems, a strict participant screening process was employed. All participants were required to have prior exposure to FRP systems and possess a certain level of understanding before completing the survey. To maintain clarity, **Table 2** presents demographic categories reflecting this sampling approach.

A pilot study was conducted prior to the full survey to assess the feasibility and content validity of the questionnaire. The primary aim of this pilot was to identify potential issues and make necessary adjustments through a small-scale pretest. Several concerns were raised, especially regarding the clarity of some questions, which had the potential to confuse respondents. As a result, adjustments were made to simplify and clarify the language, ensuring better alignment with the cognitive levels of older respondents. These revisions enhanced the clarity of the questionnaire, thus improving the accuracy and reliability of the data collected. The offline component of the survey was conducted by trained research staff to ensure that elderly participants understood each question and could independently complete the survey. Additionally, all participants were required to sign an informed consent form prior to the survey, which clearly outlined the study's purpose, content, and requirements. This informed consent form also included the contact details of research assistants for any necessary explanations, ensuring that participants could easily seek clarification if needed. This study has been approved by the Biomedical Ethics Committee of Hefei University of Technology, with the ethical approval number [HFUT20240620001H]. The online survey was conducted via the Chinese data collection platform Sojump (www.wjx.cn) between July and August 2024. Elderly participants who completed the survey received a compensation of 5 RMB each. A total of 450 responses were collected. Rigorous data screening was applied, including attention checks, reverse questions, and an assessment of survey completion times (excluding surveys completed in under 5 minutes). As a result, 387 fully completed questionnaires were retained for further analysis. The demographic information obtained from the first section of the questionnaire is provided in **Table 2**. The study does not involve any ethical concerns. Prior to participation, all participants were informed about the purpose, content, and their rights regarding the study, with explicit consent obtained for voluntary participation. During the data collection process, all participant information was kept strictly confidential, and the survey results were used solely for academic research, ensuring no personal identification information was disclosed.

**Table 2. Summary of participant demographics.**

| Variable | Description | Count | Percentage |
|---|---|---|---|
| **Gender** | Male | 185 | 47.8 |
| | Female | 202 | 52.2 |
| **Age** | 55-60 years | 254 | 65.6 |
| | 60-64 years | 46 | 11.9 |
| | 65-69 years | 40 | 10.3 |
| | 70 years and above | 47 | 12.1 |
| **Education** | Junior high school | 195 | 50.4 |
| | High School | 75 | 19.4 |
| | Junior college education | 53 | 13.7 |
| | College/university | 48 | 12.4 |
| | Postgraduate | 16 | 4.1 |
| **Income** | Less than2000 RMB | 150 | 38.8 |
| | 2000-3500 RMB | 78 | 20.2 |
| | 3500-5000 RMB | 84 | 21.7 |
| | Above 5000 RMB | 75 | 19.4 |
| **Time of using FRP** | Less than 6 months | 76 | 19.6 |
| | 6 months–1 year | 49 | 12.7 |
| | 1~3years | 58 | 15 |
| | More than 3 years | 31 | 8 |
| | Never used | 173 | 44.7 |

Moreover, the study did not involve any activities or interventions that could potentially cause physical or psychological discomfort to the participants.

## 4.2. Measurements

To assess the research constructs in this study, validated multi-item scales, commonly used in previous research [13,20,72], were employed. Each variable was observed using 3–4 scale items, aiming to measure the variable precisely from different dimensions. The specific scale items are detailed in Table 3. All items were rated using a 7-point Likert scale, ranging from 1 (strongly disagree) to 7 (strongly agree). Prior to the survey administration, considerable efforts were invested in developing, testing, and refining the items, covering aspects such as question formulation, response options, and scale anchors, to ensure the clarity and accuracy of the survey items. The reliability and validity of the measurement items were evaluated using Cronbach's alpha and confirmatory factor analysis (CFA).

## 5. Date analysis and findings

### 5.1. Analytical strategy and approach

This study employed Structural Equation Modeling (SEM) to examine the hypothesized relationships and pathways within the research framework. SEM, a robust statistical methodology with a history spanning nearly a century, integrates Confirmatory Factor Analysis (CFA) and path analysis to explore complex multivariate causal relationships. Rooted in psychometrics, CFA is designed to measure latent psychological constructs such as attitudes and satisfaction [156], making it particularly well-suited for investigating factors influencing older adults' acceptance of Facial Recognition Payment (FRP) systems. Preliminary analyses were conducted using SPSS 23.0 for descriptive statistics and reliability assessments. Confirmatory factor analysis (CFA) and evaluation of the measurement model, including path analysis, were performed

**Table 3. Questionnaire items and corresponding references.**

| Construct | Item | Details | References |
|---|---|---|---|
| **Perceived Ease of Use (PEOU)** | PEOU1 | I find using facial recognition payment straightforward and convenient. | [13] |
| | PEOU2 | The use of facial recognition payment is well within my capabilities. | [72] |
| | PEOU3 | Learning to use facial recognition payment does not pose any significant challenges for me. | [76] |
| **Perceived Usefulness (PU)** | PU1 | Facial recognition payment facilitates quick transactions. | [13] |
| | PU2 | Using facial recognition payment is beneficial. | [20] |
| | PU3 | Facial recognition payment positively impacts my daily activities. | |
| **Attitude (ATT)** | ATT1 | Using facial recognition payment is a favorable choice. | [71] |
| | ATT2 | Adopting facial recognition payment is a prudent decision. | [72] |
| | ATT3 | I prefer using facial recognition payment for purchases. | |
| **Behavioral Intention (BI)** | BI1 | I plan to use facial recognition payment in the coming months. | [71] |
| | BI2 | I intend to continue using facial recognition payment for purchases. | [72] |
| | BI3 | Overall, I am inclined to adopt facial recognition payment for purchasing goods. | |
| **Perceived convenience (PC)** | PC1 | Using facial recognition for payments simplifies the shopping experience. | [82] |
| | PC2 | The convenience of not having to manually operate when using facial recognition payment is notable. | [83] |
| | PC3 | Facial recognition payment has significantly enhanced the convenience of my daily life. | |
| **Perceived Enjoyment (PE)** | PE1 | Using facial recognition payment provides me with considerable enjoyment. | [13] |
| | PE2 | I find that using facial recognition payment enhances my shopping experience. | |
| | PE3 | The process of using facial recognition payment is engaging. | |
| **Personal Innovation (PI)** | PI1 | When I learn about a new technology, I am eager to try it. | [13] |
| | PI2 | I am known among my friends for being keen to experiment with new technologies. | [98] |
| | PI3 | Experimenting with various technologies is enjoyable. | |
| **Technological Self-efficacy (TSE)** | TSE1 | I am confident in my ability to acquire skills related to technologies (e.g., facial recognition). | [154] |
| | TSE2 | I am capable of keeping up with significant technological advancements. | [155] |
| | TSE3 | If someone demonstrates how to use a technology, I can use it to accomplish a task. | |
| | TSE4 | With a user manual, I am able to use the technology to complete a task. | |
| **Facilitating Conditions (FC)** | FC1 | I can easily obtain information about how to operate facial recognition payment systems. | [13] |
| | FC2 | Facial recognition payment integrates seamlessly with other technologies I frequently use (e.g., WeChat Pay, Alipay). | [98] |
| | FC3 | My family and friends believe/support that I should use facial recognition payment. | [154] |
| | FC4 | I have the flexibility to use facial recognition payment whenever it is necessary or desirable. | |
| **Technology Anxiety (TA)** | TA1 | Utilizing facial recognition payment induces a sense of anxiety in me. | [141] |
| | TA2 | The act of using facial recognition payment makes me feel quite uneasy. | [142] |
| | TA3 | Using facial recognition payment makes me feel uncomfortable. | |
| | TA4 | Facial recognition payment induces feelings of unease and confusion. | |

*(Continued)*

**Table 3.** (Continued)

| Construct | Item | Details | References |
|---|---|---|---|
| **Privacy Risk (PR)** | SP1 | I fear that my personal information might be exposed when using facial recognition payment. | [71] [73] [75] |
| | SP2 | I find it troubling when I am unsure of what information facial recognition payment will record. | |
| | SP3 | I am concerned that facial recognition payment gathers excessive information about me. | |
| | SP4 | I am apprehensive about conducting financial transactions via facial recognition payment. | |

with AMOS 24.0 [157]. SEM, leveraging maximum likelihood estimation, enables a holistic examination of the proposed model and provides a comprehensive analysis of inter-variable relationships based on empirical data [158]. The analytical process adhered to a two-step approach [159]: initially, CFA was conducted to validate the measurement model, followed by SEM to evaluate the structural model's fit and test the hypothesized path coefficients. Acknowledging the sensitivity of the Chi-square test to sample size, Brown [160] advocated the use of complementary fit indices to achieve a more robust evaluation of model fit. Accordingly, this study incorporated metrics such as the Comparative Fit Index (CFI), Bollen's Incremental Fit Index (IFI), Tucker-Lewis Index (TLI), Standardized Root Mean Square Residual (SRMR), and Root Mean Square Error of Approximation (RMSEA) to comprehensively assess the overall model fit. These metrics were instrumental in providing a comprehensive understanding of the model's adequacy.

## 5.2. Findings

To evaluate and confirm the causal pathways among the latent variables presented in Fig 1, structural equation modeling (SEM) was utilized. In this study, fit indices such as CFI, IFI, TLI, RMSEA, and SRMR were employed due to their proven effectiveness in past simulations [161,162]. These indices together assess the adequacy of the model, with the recommended reference values for each presented in **Table 4**. Specifically, CFI, IFI, and TLI values, averaging around 0.93, highlight a robust fit, while RMSEA and SRMR values below 0.071 indicate satisfactory alignment with the data. The calculated fit indices are as follows: $\chi^2/df = 2.942$, CFI = 0.933, IFI = 0.933, TLI = 0.924, SRMR = 0.0279, and RMSEA = 0.071. To ensure the reliability of the measurement instruments, Cronbach's alpha coefficients were computed, with all values surpassing the 0.7 threshold. The internal consistency of the scales utilized is thereby confirmed [163]. Moreover, a confirmatory factor analysis (CFA) was performed to evaluate the constructs' internal consistency, convergent validity, and discriminant validity (refer to **Table 5**). The analysis revealed that the composite reliability (CR) values ranged from 0.882 to 0.937, which exceeds the threshold of 0.60, indicating robust internal consistency [164,165]. All item factor loadings were statistically significant (p < 0.001), providing strong evidence for the convergent validity of the measurement model. Additionally, the average variance extracted (AVE) values, ranging from 0.723 to 0.876, surpassed the minimum threshold of 0.50, offering further support for convergent validity. To assess discriminant validity, the square root of the AVE for each construct was compared against the correlations among latent variables. As presented in **Table 6**, diagonal elements

**Table 4. Model fit indices.**

| Fit indices | $\chi^2/df$ | CFI | IFI | TLI | SRMR | RMSEA |
|---|---|---|---|---|---|---|
| **Reference value** | <3.000 | >0.900 | >0.900 | >0.900 | <0.080 | <0.080 |
| **Inspection value** | 2.942 | 0.933 | 0.933 | 0.924 | 0.0279 | 0.071 |

**Table 5. Reliability and unidimensionality assessment.**

| Construct | Variables | Unstd. | S.E. | Z | P | Std. | Cronbach's Alpha | C.R. | AVE |
|---|---|---|---|---|---|---|---|---|---|
| PEOU | PEOU1 | 1.000 | | | | 0.892 | 0.932 | 0.933 | 0.823 |
| | PEOU2 | 1.012 | 0.036 | 27.974 | *** | 0.914 | | | |
| | PEOU3 | 1.000 | 0.036 | 28.049 | *** | 0.915 | | | |
| PU | PU1 | 1.000 | | | | 0.929 | 0.923 | 0.924 | 0.802 |
| | PU2 | 0.980 | 0.033 | 30.125 | *** | 0.899 | | | |
| | PU3 | 0.922 | 0.035 | 26.481 | *** | 0.858 | | | |
| ATT | ATT1 | 1.000 | | | | 0.892 | 0.929 | 0.927 | 0.810 |
| | ATT2 | 0.999 | 0.038 | 26.291 | *** | 0.882 | | | |
| | ATT3 | 1.107 | 0.037 | 29.654 | *** | 0.925 | | | |
| BI | BI1 | 1.000 | | | | 0.894 | 0.954 | 0.955 | 0.876 |
| | BI2 | 1.041 | 0.032 | 32.512 | *** | 0.954 | | | |
| | BI3 | 1.061 | 0.032 | 32.908 | *** | 0.958 | | | |
| PC | PC1 | 1.000 | | | | 0.930 | 0.929 | 0.929 | 0.815 |
| | PC2 | 0.959 | 0.031 | 30.633 | *** | 0.903 | | | |
| | PC3 | 0.944 | 0.034 | 27.857 | *** | 0.874 | | | |
| PE | PE1 | 1.000 | | | | 0.912 | 0.941 | 0.941 | 0.842 |
| | PE2 | 1.041 | 0.033 | 31.747 | *** | 0.937 | | | |
| | PE3 | 0.997 | 0.035 | 28.718 | *** | 0.903 | | | |
| PI | PI1 | 1.000 | | | | 0.893 | 0.894 | 0.894 | 0.737 |
| | PI2 | 0.998 | 0.041 | 24.205 | *** | 0.875 | | | |
| | PI3 | 0.928 | 0.045 | 20.688 | *** | 0.805 | | | |
| TSE | TSE1 | 1.000 | | | | 0.862 | 0.930 | 0.929 | 0.767 |
| | TSE2 | 0.957 | 0.042 | 22.641 | *** | 0.856 | | | |
| | TSE3 | 1.029 | 0.042 | 24.672 | *** | 0.894 | | | |
| | TSE4 | 1.010 | 0.041 | 24.479 | *** | 0.890 | | | |
| FC | FC1 | 1.000 | | | | 0.851 | 0.912 | 0.912 | 0.723 |
| | FC2 | 0.954 | 0.046 | 20.547 | *** | 0.826 | | | |
| | FC3 | 1.003 | 0.046 | 21.681 | *** | 0.852 | | | |
| | FC4 | 1.020 | 0.045 | 22.577 | *** | 0.871 | | | |
| TA | TA1 | 1.000 | | | | 0.869 | 0.918 | 0.919 | 0.741 |
| | TA2 | 1.026 | 0.043 | 24.056 | *** | 0.893 | | | |
| | TA3 | 1.018 | 0.045 | 22.376 | *** | 0.858 | | | |
| | TA4 | 0.968 | 0.047 | 20.694 | *** | 0.821 | | | |
| PR | PR1 | 1.000 | | | | 0.860 | 0.929 | 0.930 | 0.770 |
| | PR2 | 1.042 | 0.042 | 24.665 | *** | 0.908 | | | |
| | PR3 | 0.993 | 0.044 | 22.331 | *** | 0.860 | | | |
| | PR4 | 1.072 | 0.046 | 23.263 | *** | 0.880 | | | |

(representing AVE square roots) were consistently higher than off-diagonal correlations, affirming the uniqueness of each construct [166].

The path analysis conducted using structural equation modeling (SEM) reveals that 10 out of the 14 hypotheses proposed in this study were strongly supported ($p < 0.001$), while 3 hypotheses received partial support ($p < 0.05$). However, the H10 hypothesis was not supported. The detailed results are presented in **Fig 2** and **Table 7**. As anticipated, perceived ease of use (PEOU) exerted a significant positive effect on perceived usefulness (PU) ($\beta = 0.366$, $p < 0.001$), and the

**Table 6. Correlations and average variance extracted.**

|  | BI | PEOU | ATT | PU | TSE | PI | PR | TA | FC | PE | PC |
|---|---|---|---|---|---|---|---|---|---|---|---|
| BI | **0.936** | | | | | | | | | | |
| PEOU | 0.801 | **0.907** | | | | | | | | | |
| ATT | 0.973 | 0.855 | **0.900** | | | | | | | | |
| PU | 0.886 | 0.847 | 0.933 | **0.896** | | | | | | | |
| TSE | 0.791 | 0.923 | 0.840 | 0.859 | **0.876** | | | | | | |
| PI | 0.749 | 0.803 | 0.780 | 0.780 | 0.887 | **0.859** | | | | | |
| PR | −0.113 | 0.011 | −0.122 | −0.015 | 0.039 | 0.048 | **0.877** | | | | |
| TA | −0.139 | −0.078 | −0.147 | −0.116 | −0.032 | 0.055 | 0.787 | **0.861** | | | |
| FC | 0.830 | 0.795 | 0.843 | 0.863 | 0.843 | 0.755 | −0.077 | −0.101 | **0.850** | | |
| PE | 0.793 | 0.680 | 0.824 | 0.795 | 0.723 | 0.713 | −0.068 | −0.070 | 0.892 | **0.917** | |
| PC | 0.830 | 0.794 | 0.896 | 0.944 | 0.880 | 0.834 | 0.016 | −0.091 | 0.856 | 0.798 | **0.903** |

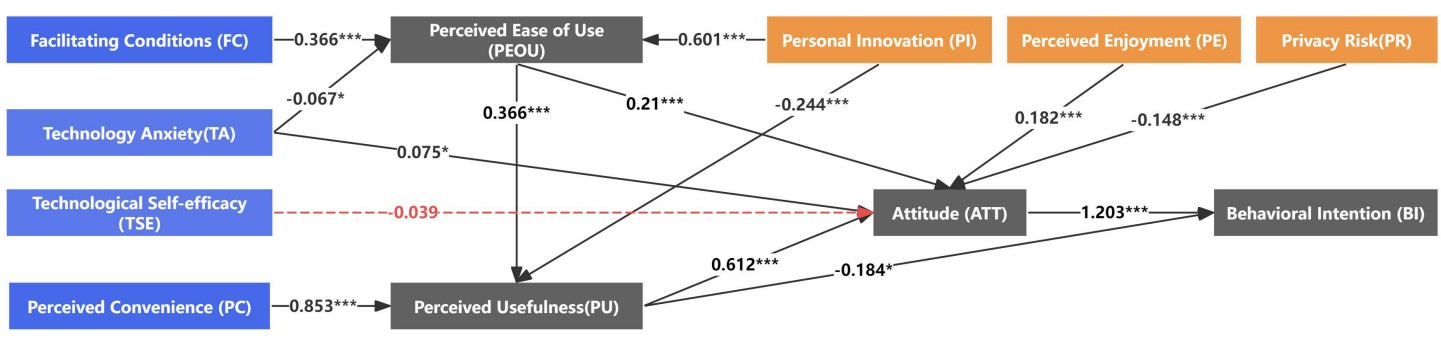

**Fig 2. Structural equation modeling results. * p<0.1; ** p<0.05; *** p<0.01.**

relationship between PU and behavioral intention (BI) was validated (β=−0.184, p<0.05), thereby confirming hypotheses H1 and H2. Elderly users' attitudes toward FRP were significantly influenced by PEOU (β=0.21, p<0.001), PU (β=0.612, p<0.001), perceived enjoyment (PE) (β=0.182, p<0.001), technology anxiety (TA) (β=0.075, p<0.05), and privacy risk (PR) (β=−0.148, p<0.001), thus affirming hypotheses H3, H4, H7, H13, and H14. Conversely, technology self-efficacy (TSE) did not have a significant effect on user attitude (ATT) (β=−0.039, p>0.1), resulting in a lack of support for H10. Furthermore, user attitude (ATT) had a positive and significant influence on behavioral intention (BI) (β=1.203, p<0.001), supporting H5. Perceived convenience (PC) was found to have a substantial positive effect on perceived usefulness (PU) (β=0.853, p<0.001), confirming H6. Personal innovation (PI) had a significant positive impact on both perceived ease of use (PEOU) (β=0.601, p<0.001) and perceived usefulness (PU) (β=−0.244, p<0.001), validating H8 and H9. Facilitating conditions (FC) (β=0.366, p<0.001) and technology anxiety (TA) (β=−0.067, p<0.05) both influenced perceived ease of use (PEOU), confirming H11 and H12.

# 6.Discussion

## 6.1. Theoretical implications

This study investigates the acceptance and usage of facial recognition payment (FRP) technology among elderly users by integrating the Technology Acceptance Model (TAM) and User Experience Theory (UXT). The results demonstrate that

**Table 7. Results of hypothesis testing.**

| | Unstd. | S.E. | Z | P-Values | Std. | Resutls |
|---|---|---|---|---|---|---|
| **H1:PEOU→PU** | 0.366 | 0.055 | 6.654 | *** | 0.358 | Supported |
| **H2:PU→BI** | −0.184 | 0.089 | −2.067 | 0.039* | −0.179 | Partially supported |
| **H3:PEOU→ATT** | 0.21 | 0.052 | 4.008 | *** | 0.212 | Supported |
| **H4:PU→ATT** | 0.612 | 0.062 | 9.836 | *** | 0.632 | Supported |
| **H5:ATT→BI** | 1.203 | 0.1 | 11.99 | *** | 1.137 | Supported |
| **H6:PC→PU** | 0.853 | 0.059 | 14.531 | *** | 0.857 | Supported |
| **H7:PE→ATT** | 0.182 | 0.036 | 5.006 | *** | 0.195 | Supported |
| **H8:PI→PEOU** | 0.601 | 0.061 | 9.911 | *** | 0.577 | Supported |
| **H9:PI→PU** | −0.244 | 0.068 | −3.584 | *** | −0.229 | Supported |
| **H10:TSE→ATT** | −0.039 | 0.052 | −0.747 | 0.455 | −0.037 | Not supported |
| **H11:FC→PEOU** | 0.366 | 0.057 | 6.369 | *** | 0.359 | Supported |
| **H12:TA→PEOU** | −0.067 | 0.031 | −2.189 | 0.029* | −0.069 | Partially supported |
| **H13:TA→ATT** | 0.075 | 0.037 | 1.998 | 0.046* | 0.078 | Partially supported |
| **H14:PR→ATT** | −0.148 | 0.039 | −3.786 | *** | −0.15 | Supported |

**\*\*\* p<0.001; \*\* p<0.01; \* p<0.05**

both Perceived Usefulness (PU) and Perceived Ease of Use (PEOU) have a substantial impact on customer attitudes, which in turn positively influence usage intentions. These findings align with the original TAM model [47].

The study reveals that, from the perspective of the Technology Acceptance Model (TAM), Perceived Convenience (PC) exerts a significant influence on Perceived Usefulness (PU) with a β value of 0.853. This indicates that Perceived Convenience has a notable effect on older users' perception of the technology's usefulness. The findings confirm that older adults are more likely to view a technology as useful if it is highly convenient to use, which subsequently influences their Behavioral Intention (BI) to adopt the technology. This underscores the critical importance of simplifying technology operations and interface design for older users. Given that older adults encounter challenges such as complex operations and steep learning curves when using new technologies, convenience becomes a key determinant of technology acceptance. This result is consistent with previous studies, which highlight the pivotal role of convenience in enhancing user experience and perceptions of mobile payment systems [83]. For instance, Facial Recognition Payment (FRP) systems do not require users to enter passwords or carry phones, with transactions completed in under 15 seconds. This level of convenience positions FRP as superior to other payment methods. Additionally, the research confirms that Facilitating Conditions (FC), as an important external variable, has a statistically significant impact on Perceived Ease of Use (PEOU). This finding underscores the importance of providing adequate resources and support for older users, as such provisions can mitigate the challenges of technology usage, enhance its effectiveness, and directly influence the intention to adopt the technology. These findings align with research by Mirjana et al., which demonstrates a significant positive relationship between social influence, social support, and older adults' self-efficacy [167]. Additionally, Chu et al. found that older adults rely more on tangible family support to improve their internet learning outcomes compared to middle-aged individuals [168]. Thus, Facilitating Conditions (FC) is a crucial factor influencing the adoption of new technologies among older users. Furthermore, the study confirms that Technological Anxiety (TA) has a negative impact on Perceived Usefulness (PU) and Attitude (ATT), consistent with previous findings [141,169]. For example, McFarland and Hamilton demonstrated that computer anxiety markedly impedes users' intentions to utilize computers [141]. Due to age-related physiological decline [38], older adults generally exhibit lower adaptability to new technologies compared to younger individuals [41,42].

This anxiety often leads to perceptions of greater complexity and operational difficulty, thereby diminishing their willingness to adopt new technologies.

The fourth variable derived from the TAM framework is Technological Self-Efficacy (TSE). However, the research findings related to this variable contradict the expected hypothesis, which is at odds with the majority of existing studies [170,171]. However, this observation aligns with the findings of Mohammed et al., who reported that General Self-Efficacy (GSE) and Computer Self-Efficacy (CSE) had minimal impact on attitudes toward health technology within the context of health technology use [172]. This finding also corroborates Bandura's assertion that self-efficacy is more context-specific than a general perception and should be tailored to specific domains [173]. In the context of older adults using facial recognition payment (FRP) technology, the lack of a significant influence of TSE on their technology use attitudes is particularly noteworthy. One possible explanation is that FRP, by employing a "face scan to pay" mechanism, significantly simplifies both the user interface and operational procedures, thereby reducing the need for users to rely on their technological self-efficacy. Unlike traditional digital payment systems, which require multiple steps such as inputting passwords or scanning QR codes, FRP operates in an almost automatic manner. This means that even individuals with lower confidence in their technological abilities can use it with ease, as the intuitive and effortless nature of FRP mitigates the need for self-efficacy in shaping user attitudes. From a User Experience Theory (UXT) perspective, these results highlight a critical insight into elderly users' technology adoption behavior. Unlike younger users, who may develop a strong attitude toward technology based on their confidence in handling digital tools, elderly users tend to evaluate new technology primarily based on its perceived simplicity, convenience, and security rather than their own technological competence. This underscores the significance of age-friendly design, which can enhance user experience by focusing on the intuitive usability of the technology and its emotional appeal. Age-friendly design reduces the cognitive and operational burden associated with new technologies, making them more acceptable to older populations. Therefore, developers should prioritize optimizing the usability of FRP systems, ensuring minimal cognitive effort and promoting seamless user experiences to foster broader adoption among elderly users.

Another perspective of this study is developed based on User Experience Theory (UXT). The findings reveal that Personal Innovation (PI) significantly impacts both Perceived Ease of Use (PEOU) and Perceived Usefulness (PU), corroborating existing literature [100]. This confirms that individuals with higher levels of personal innovation tend to have a more favorable view of the usefulness of new technologies [104]. This effect is especially pronounced among older adults, who are generally more inclined to experiment with and adapt to new technologies when they exhibit higher personal innovation, as also evidenced by research on older users' acceptance of voice assistants [174]. This insight suggests that incorporating strategies to stimulate and enhance personal innovation(PI) could significantly improve technology adoption rates and user experience for older adults.

The second variable derived from User Experience Theory (UXT) is Perceived Enjoyment (PE), the study finds that Perceived Enjoyment directly influences older users' attitudes towards FRP, validating that the enjoyment derived from using a technology is crucial for its acceptance [84]. This finding further substantiates the pivotal role of Emotional Design in enhancing user experience. Integrating interactive feedback, engaging elements, and personalized customization tailored to older users within interface design can significantly improve their overall interaction and satisfaction. Furthermore, this aligns with the findings of Hornbæk and Hertzum, who highlighted that user experience exerts a more substantial influence on technology acceptance than perceived usefulness and ease of use, particularly through the impact of perceived enjoyment on user attitudes [35]. These results underscore that, beyond technical factors, emotional and cognitive aspects play a critical role in shaping technology adoption among older users [36].

In addition to Personal Innovation (PI) and Perceived Enjoyment (PE), the third variable derived from User Experience Theory (UXT) is Privacy Risk (PR). The findings demonstrate that Privacy Risk (PR) significantly negatively impacts older users' attitudes toward Facial Recognition Payment (FRP), which is consistent with prior research [11,21,151]. This indicates that concerns about privacy often drive users to resist adopting FRP technology. Recent studies further highlight

that privacy risk is a critical factor contributing to user reluctance toward FRP technology, particularly when users perceive a high level of privacy risk [152]. Privacy risk is particularly pronounced among older users, who tend to be more sensitive to privacy issues associated with emerging technologies. Specifically, technologies involving the processing of personal data, such as FRP systems that require the upload or storage of biometric information, can evoke considerable unease and concern in older individuals. This discomfort is largely attributed to fears of privacy breaches and a lack of trust in the technology's protective measures, which may lead to reduced acceptance of FRP technology and negatively affect their willingness to adopt it. Furthermore, the relatively recent implementation of privacy-related legislation in China may contribute to a less mature level of privacy awareness and understanding among older citizens [175]. This context further elucidates the heightened concern regarding privacy risks among older users. The user experience theory places substantial emphasis on the comprehensive subjective perceptions that users develop regarding products or services. Cognition, within this context, occupies a pivotal position. In line with cognitive psychology theory, during the decision-making process, humans typically perform risk assessments of novel entities by leveraging their pre - established cognitive patterns. The elderly, being fettered by traditional mindsets and possessing relatively restricted technological knowledge, are more inclined to assume a cautious stance when confronted with new technologies compared to other demographic groups. In the case of Facial Recognition Payment (FRP), once the elderly become aware that it has the potential to result in severe outcomes such as privacy infringement, identity theft, or property damage, this substantially diminishes their propensity to use FRP [27,28].

## 6.2. Practical implications

The findings of this study have significant practical implications for the design and implementation of age-friendly Facial Recognition Payment (FRP) systems and analogous technologies tailored for older users. The results reveal that Perceived Convenience (PC) significantly influences Perceived Usefulness (PU), with a β coefficient of 0.853. This indicates that older users are more likely to perceive technologies that are easy to use as more useful. Therefore, developers should adopt the "Less is More" principle in UX design by streamlining payment processes, offering intuitive operational guidance, and integrating real-time feedback mechanisms to optimize the user experience [176]. Additionally, the study highlights that Perceived Enjoyment (PE) directly impacts older users' attitudes towards FRP. This underscores the importance of incorporating elements of entertainment and pleasant experiences into technology design for older users. Design teams should integrate interactive and personalized features into FRP systems to enhance user engagement and enjoyment, thereby improving the overall user experience.

Moreover, Facilitating Conditions (FC) significantly affect Perceived Ease of Use (PEOU), emphasizing the critical role of adequate technical support and resources in the adoption of technology by older users. Therefore, government agencies should consider implementing measures such as providing detailed operational guides, organizing regular technology training sessions for older users, and establishing technical support hotlines [177]. The negative impact of Technological Anxiety (TA) should also be addressed. Older adults frequently experience anxiety towards new technologies, which can increase perceived complexity and reduce adoption willingness. Developers should focus on mitigating this anxiety by offering clear instructions, conducting training sessions, and designing intuitive systems to enhance technology acceptance. Although Technological Self-Efficacy (TSE) did not significantly affect older users' technology attitudes in this study, it remains important to enhance users' confidence. Future research should explore how self-efficacy interacts with other factors, such as technology design and environmental support, across various technological contexts. This approach will contribute to a more nuanced understanding of older adults' technology use behaviors and aid in optimizing technology design. By integrating these considerations, we can enhance technology usability and acceptance, ultimately creating a more user-friendly technological environment for older adults.

Lastly, the study identifies Privacy Risk as a negative factor influencing older users' acceptance of FRP technology. This finding provides crucial guidance for both technology developers and policymakers. FRP service providers should

ensure that privacy protection measures are transparent and reliable to build older users' trust in the technology's security. Designers and developers should also address privacy protection needs by incorporating privacy control options and clear privacy management features into the technology interfaces. This approach can help mitigate older users' concerns about data security and enhance technology acceptance. Furthermore, given the relatively recent implementation of privacy-related laws in China, it is recommended that the government strengthen public education on privacy protection, increase awareness among older users, and continuously refine privacy regulations to adapt to evolving technological landscapes and user needs, thus effectively safeguarding users' privacy rights.

## 7. Conclusion and limitations

This study offers several significant theoretical contributions to the academic community. Firstly, unlike existing research that predominantly focuses on general or younger populations, this study uniquely investigates the usage and acceptance of Facial Recognition Payment (FRP) among elderly users in China. Given the dual challenges of aging and digitalization in contemporary society, technologies that support and benefit the elderly represent an essential trend in social development. The "face scan to pay" feature of FRP largely mitigates the "digital divide" for older users. The primary contribution of this research lies in exploring the factors and mechanisms affecting the adoption of FRP among elderly users, promoting the technology within this demographic, and enhancing their shopping experience and overall quality of life. Secondly, in the realm of theoretical model construction, this research demonstrates remarkable innovation by seamlessly integrating the Technology Acceptance Model (TAM) and the User Experience Technology (UXT) framework. A comprehensive and highly tailored technology acceptance integrated model for the elderly user segment is meticulously formulated. From the TAM vantage point, this study pioneers the incorporation of four external variables, namely Perceived Convenience (PC), Facilitating Conditions (FC), Technological Anxiety (TA), and Technological Self-Efficacy (TSE). This endeavor successfully surmounts the inherent constraints of the traditional TAM model in elucidating the technology acceptance behaviors of elderly users. As a result, it significantly broadens the application frontier of the TAM model within the purview of specific user-group research, thus charting a novel course for in-depth exploration of the technology adoption behaviors of the elderly. In the context of the UXT perspective, this research introduces three external variables: Perceived Enjoyment (PE), Personal Innovation (PI), and Privacy Risk (PR). This approach comprehensively takes into account the emotional experiences and distinctive psychological traits of elderly users during the technology utilization process. It effectively remedies the deficiency in the attention to the emotional dimensions of elderly users in prior related research, rendering the study of the technology acceptance behaviors of the elderly more comprehensive and multi – faceted, thereby validating and extending previous research [13,71–76,83].

Through rigorous and scientific empirical investigations, this study conclusively validates the existence of a close and significant correlation between these newly introduced variables and the acceptance level of Facial Recognition Payment (FRP) among elderly users. This accomplishment not only furnishes a more consummate and precise theoretical analysis instrument for subsequent research but also substantially extends the TAM from a novel perspective that focuses on the hierarchical structure of user needs. This will powerfully stimulate more scholars to conduct in-depth inquiries into the technology acceptance behaviors of elderly users from multi-dimensional and cross - framework perspectives. Consequently, it will continuously propel the refinement and development of the theoretical system in this domain, injecting new impetus and orientation into the research and practice of technology applications relevant to the elderly population.

The findings provide crucial insights for the design and development of age-friendly FRP systems. Firstly, designers should adhere to the "user-centered" principle inherent in user experience theory, simplifying the FRP usage process, while emphasizing user engagement and the enjoyment experienced during the interaction, thereby enhancing the overall user experience. Secondly, drawing from both TAM and UXT, it is crucial to provide adequate technical support and assistance mechanisms to foster elderly users' technological self-efficacy, which remains a pivotal focus for product designers and developers. Lastly, developers and policymakers must consider privacy concerns by incorporating privacy protection

options into technology interfaces and strengthening public education on privacy protection to safeguard users' privacy rights effectively. Our findings are anticipated to motivate future research efforts aimed at advancing the age-friendliness and user experience of facial recognition payment (FRP) systems. These improvements hold the potential to foster greater well-being and independence among older adults, thereby enhancing their overall quality of life.

Although this study provides valuable insights and practical implications, certain limitations should be acknowledged. One primary limitation is the sample, which is restricted to elderly participants exclusively from China. FRP services may vary across different countries and cultures, and technology maturity and market conditions differ internationally. Therefore, future research should encompass elderly users from various countries and cultural backgrounds to enhance the generalizability of the current model. Additionally, the study found that Technological Self-Efficacy did not significantly impact Attitude, despite its recognized importance in promoting and stimulating the adoption of new technologies among elderly users. This warrants further analysis. As Bandura posited, self-efficacy is highly context-specific and should be tailored to particular domains [173]. Future studies should explore how self-efficacy interacts with other factors, such as technology design and environmental support, across diverse technological contexts to deepen the understanding of elderly users' technology usage behaviors. Finally, while the combined TAM and UXT research model incorporates seven external variables, future studies should delve deeper into additional factors, such as emotional design and usability testing, which hold significant potential for further exploration within the context of TAM and UXT.

## Supporting information

**S1 Data. The original survey data of this study.**
(XLSX)

## Author contributions

**Conceptualization:** Caixia WANG.

**Data curation:** Caixia WANG, Jing HU.

**Formal analysis:** Mingzhou LIU, Caixia WANG, Jing HU.

**Investigation:** Caixia WANG, Jing HU.

**Methodology:** Caixia WANG, Jing HU.

**Project administration:** Mingzhou LIU.

**Resources:** Caixia WANG.

**Software:** Caixia WANG, Jing HU.

**Supervision:** Jing HU.

**Validation:** Mingzhou LIU, Jing HU.

**Visualization:** Caixia WANG.

**Writing – original draft:** Caixia WANG.

**Writing – review & editing:** Mingzhou LIU, Caixia WANG, Jing HU.

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
