## [Decision Letter · Decision Letter 0]

PONE-D-24-56273Are Facial Recognition Payments Aging-Friendly? Investigating Acceptance Through an Extended Technology Acceptance ModelPLOS ONE?

Dear Dr. LIU,

Thank you for submitting your manuscript to PLOS ONE. Please provide detailed responses to each of the reviewers' comments. Additionally, proofreading is necessary to address any existing typos and grammatical errors. As per PLOS' policy, authors are required to disclose their research data. Since your data is not proprietary, it must be made publicly available.

We look forward to receiving your revised manuscript.

Kind regards,

Simon Dang

Academic Editor

PLOS ONE

Journal Requirements:

Additional Editor Comments:

Reviewers' comments:

Reviewer's Responses to Questions

**Comments to the Author**

1. Is the manuscript technically sound, and do the data support the conclusions?

Reviewer #1: Partly

Reviewer #2: Yes

2. Has the statistical analysis been performed appropriately and rigorously?

Reviewer #1: Yes

Reviewer #2: Yes

3. Have the authors made all data underlying the findings in their manuscript fully available?

Reviewer #1: Yes

Reviewer #2: No

4. Is the manuscript presented in an intelligible fashion and written in standard English?

Reviewer #1: Yes

Reviewer #2: Yes

Reviewer #1: Are Facial Recognition Payments Aging-Friendly? Investigating Acceptance Through an Extended Technology Acceptance Model

1. The main issue with this study is its lack of originality. While the study's focus on facial recognition payment in the field of elderly populations is meaningful, I believe it is somewhat insufficient for publication. Therefore, it is better to emphasize the originality in the Introduction section.

2. The study employs Technology Acceptance Model, but it is worth considering other theories in order to support the research model in the study.

3. It would be beneficial to add empirical studies for hypotheses 6 to 14

4. Citing the following paper would help enhance the quality of the theoretical background in this study.

Joo et al. (2024). Effects of foodservice consumers’ perceptions of face recognition payment on attitude, desire, and behavioral intentions: a cross-cultural study. Journal of Travel & Tourism Marketing, 41(3), 359-376.

Hwang et al. (2024). Effects of motivated consumer innovativeness on facial recognition payment adoption in the restaurant industry: A cross-cultural study. International Journal of Hospitality Management, 117, 103646.

Kim et al. (2024). The antecedents and consequences of task–technology fit of facial recognition payment systems in the restaurant industry: cultural differences. Journal of Hospitality and Tourism Technology, 15(3), 397-416.

Hwang et al. (2024). An integrated model of artificially intelligent (AI) facial recognition technology adoption based on perceived risk theory and extended TPB: A comparative analysis of US and South Korea. Journal of Hospitality Marketing & Management, 33(8), 1071-1099.

5. All measurement items should be presented.

6. A discussion is also necessary for hypotheses that are not statistically significant.

7. The theoretical implications were presented in comparison with existing research, but they do not carry significant meaning.

Reviewer #2: Thank you for giving me the opportunity to review this paper. As the author(s) mentioned, this study examined FRP, focusing on elderly consumers. This study is meaningful in that it addresses the gaps of technology research that primarily focuses on younger consumers. There are suggestions that need to be taken into consideration and to be corrected for increasing the quality of this paper.

1. This paper is well-written, and the ideas are presented in a clear manner.

2. What is the reasoning behind reporting H2, H12, and H13 as ‘partially supported?’

3. What is a logical explanation for why TSE did not have a significant effect on attitude specifically in the context of elderly consumers?

4. The paper states that the data collection focused on individuals aged 60 and above, but Table 2 lists an age group from 55-60 years old. Were all participants over the age of 60 or were the female participants following the threshold of the Chinese official retirement age?

5. Were all participants previously familiar with FRP systems or was there some type of informational materials given to the participants to help them gain an understanding of how FRP systems operate prior to completing the survey?

**Do you want your identity to be public for this peer review?** For information about this choice, including consent withdrawal, please see our Privacy Policy

Reviewer #1: No

Reviewer #2: No

---

## [Author Response · Author response to Decision Letter 1]

10 Apr 2025

Dear Editor,

We sincerely appreciate your valuable feedback and constructive suggestions regarding our manuscript.

Firstly, we have carefully reviewed all revision comments and suggestions and have conducted in-depth reflections and comprehensive revisions based on each point raised by the journal and the editors. We have thoroughly considered the theoretical innovations in our study and integrated the User Experience Theory (UXT) with the Technology Acceptance Model (TAM) to construct a comprehensive framework for examining the acceptance of facial recognition payment technology among elderly users. This enhancement has significantly improved the theoretical depth and innovation of our manuscript.

Secondly, regarding the ethical documentation required for the study, such as the informed consent form, we have supplemented the submission with the relevant materials. These have been provided under the "Other" category, including both an English version and a Chinese sample for your review.

Additionally, we are pleased to inform you that we have fully complied with the journal’s requirement for research data sharing. All data related to the research findings described in our manuscript are now publicly available without any restrictions. These data can be accessed from the public dataset on figshare. The URL of the dataset is: https://doi.org/10.6084/m9.figshare.28774835, and its Digital Object Identifier (DOI) is 10.6084/m9.figshare.28774835. We believe this open data approach will facilitate further research and validation in the field.

Furthermore, detailed revision information can be found in the system-submitted file “Response to Reviewers”.

Finally, we sincerely appreciate your guidance throughout this process. If any further clarification or additional information is needed, please do not hesitate to contact us.

Thank you again for your support!

Best regards,

[Mingzhou LIU, Caixia WANG, Jing HU]

---

## [Decision Letter · Decision Letter 1]

Understanding Older Adults’Adoption of Facial Recognition Payment: An Integrated Model of TAM and UXT

PONE-D-24-56273R1

Dear Dr. LIU,

We’re pleased to inform you that your manuscript has been judged scientifically suitable for publication and will be formally accepted for publication once it meets all outstanding technical requirements.

Kind regards,

Simon Dang, Ph.D.

Academic Editor

PLOS ONE

Additional Editor Comments (optional):

Thank you for submitting your manuscript to PLOS ONE and thank you for giving us the opportunity to continue to work with you. I am delighted to inform you that your manuscript has now been accepted for publication. This is a significant achievement, and I would like to extend my heartfelt congratulations to you. We look forward to sharing your valuable contribution with the broader academic community. Well done!

Reviewers' comments:

Reviewer's Responses to Questions

**Comments to the Author**

Reviewer #1: All comments have been addressed

Reviewer #2: (No Response)

2. Is the manuscript technically sound, and do the data support the conclusions?

Reviewer #1: Yes

Reviewer #2: Yes

3. Has the statistical analysis been performed appropriately and rigorously?

Reviewer #1: Yes

Reviewer #2: Yes

4. Have the authors made all data underlying the findings in their manuscript fully available?

Reviewer #1: Yes

Reviewer #2: Yes

5. Is the manuscript presented in an intelligible fashion and written in standard English?

Reviewer #1: Yes

Reviewer #2: Yes

Reviewer #1: First of all, I would like to thank the authors for their efforts. They sincerely revised the manuscript according to my comments, and the quality of the paper has significantly improved. Well done.

Reviewer #2: (No Response)

**Do you want your identity to be public for this peer review?** For information about this choice, including consent withdrawal, please see our Privacy Policy

Reviewer #1: No

Reviewer #2: No

---

## [Editor Report · Acceptance letter]

PONE-D-24-56273R1

PLOS ONE

Dear Dr. LIU,

I'm pleased to inform you that your manuscript has been deemed suitable for publication in PLOS ONE. Congratulations! Your manuscript is now being handed over to our production team.

Kind regards,

on behalf of

Dr. Simon Dang

Academic Editor

PLOS ONE